# Performance Characteristics of a Small Hammer Head Pump

Krishpersad Manohar[1], Anthony Ademola Adeyanju[1] and Kureem Vialva[1]
[1]Mechanical and Manufacturing Engineering Department,
The University of the West Indies,
St. Augustine, Trinidad, West Indies

*Correspondence to*: Krishpersad Manohar (krishpersad.manohar@sta.uwi.edu)

**Abstract:**
Many rural farming areas are located far from reliable electricity supply, hence, having a reliable source of water for crops and livestock can prove to be an expensive venture. A water pump operating on the water hammer effect requires no external power source and can serve as an effective means of pumping water to a higher altitude once a reliable supply is available. A low-cost small hammer-head pump was designed to operate on the hammer head effect created by the sudden stoppage of a flowing fluid. This design consisted of an inlet section followed by the pump body, a pressure section and an outlet. The experimental set-up for testing the hammer head pump was designed with a variable input and an adjustable head output. For each test configuration, ten samples of pump supply water and pump exhausted water were collected. The water samples were collected for 30s in each case. The results showed a non-linear variation of water flow respect to pump outlet height. The pump was capable of delivering water to a maximum height of 8 to 10 times the height of the input head. The pump operated at average efficiencies of 26%, 16% and 6% when the delivery height was twice, four times and six times the input head, respectively. There was a 5% incremental decrease in pump efficiency as the delivery height increased in increments of the corresponding input head height.

**Keywords:**
Hammer head, hydraulic ram. , water pump.

## Introduction

The first type of pumps to use the water hammer effect was the hydraulic ram pump which was reported in 1775 and was built by John Whitehurst [1]. His design was not automatic and was controlled by manually opening and closing a stopcock which resulted in the device only being able to raise water to a height of 4.9 meters. This involved a significant amount of work and consumed a lot of time to operate. However, in 1797 the design was improved and the first reported automatic hydraulic ram was developed by Joseph and Etienne Montgolfier to raise water to a paper mill [2]. Although this was an improved design it still contained design flaws which caused the air in the pressure chamber to dissolve or drop. In 1816 this problem was eliminated when Pierre Montgolfier designed the sniffer valve that reintroduced air into the chamber. This valve was 15 cm in radius and it was reported that the pump was able to raise water to 48 meters in height [3]. The automatic hydraulic ram has been used for centuries to lift water to heights over 100 meters and was considered an effective and highly reliable machine for pumping water once certain conditions are satisfied. The pump construction was simple and consisted of a pump chamber fitted with two moving parts, an impulse valve through which the driving water was wasted and a delivery valve through which the water was delivered [4]. It worked solely on the power supplied from the water head in the source. This source could be a spring, streams, river, ponds, dam, lakes and even some wells, once the conditions existed for these water sources to create a hydraulic flow head, either by forming a dam or a naturally existing head. Basically, once a hydraulic head can be created, the pump can operate, however, the source must provide a steady and reliable supply of water [5]. The ram pump must be installed at a location lower than the water source which was used to create the flow giving the fluid (water) some velocity.

In many rural farming areas, having a reliable source of water for crops and livestock can prove to be an expensive venture. In developing and under-developed countries, farmlands are ideally located close to a reliable water source to ensure viability [6, 7]. However, in many instances these locations are far from any reliable source of electricity and the cost can be prohibitive [6, 8]. The alternative of diesel-driven pumps create high operation costs and are prone to service gaps due to insufficient fuel supply and technical defects. A reliable and cost-effective supply of irrigation water is therefore a core problem in many rural areas in developing and emerging countries [9]. In cases where the water source is situated below the level of the farmlands, getting the water to where it is needed can be challenging [7]. Under these circumstances, a water pump operating on the water hammer effect and requires no external power source can serve as an effective means of pumping water to a higher altitude once a reliable source is available. Also, in under developed countries, such as Haiti, the feasibility of using small hammer head pumps to provide clean water for citizens were explored by Prude University [10]. The ram pump can operate 24/7 and hence a water storage facility, such as storage tanks, at the water delivery end will be needed. This will serve as the reservoir to supply the needs when required. The major hindrance in using this established technology in third world countries is the exorbitant cost of the commercially available units. For a UK built pump the cost is US$ 1800 [11] and cheaper china made pumps range between US$500 to US$1300 [12]. One of the objectives of the Prude University project in Haiti was to develop a cheaper alternative, however, the cost was US$100 [10]. Therefore, there is the need to develop a low cost alternative that can be easily built from readily available construction materials and requires minimal technical skills.

Given the long history of the hydraulic ram pump, the design and manufacture has improved considerably with time and efficiency of operation increased. For commercial ram pumps the typical energy efficiency is about 60%, but can reach up to 80% [13].This is different from the volumetric efficiency, which relates the volume of water delivered to total water taken from the source. The amount of water delivered will be reduced by the ratio of the output head to the supply head. For example, if the source is 2 meters above the ram pump and the water is lifted to 10 meters above, only 20% of the supplied water will be available and the other 80% being spilled via the exhaust valve [14]. These ratios assumed 100% energy efficiency. The actual water delivered will be reduced further by the energy efficiency. Hence, for an energy efficiency of 70%, the water delivered will be 70% of 20%, which yields14% [14, 15]. Suppliers of ram pumps often provide tables giving expected volume ratios based on actual tests. The amount of water delivered to the end for use will depend on source flow, height of supply reservoir above pump, height of delivery site above pump, length and size of delivery pipe and drive line, pump efficiency, and size of pump [15, 16, 17]. Considering the many combinations of these variables, the amount of water that can be delivered vary significantly. For example, delivery output from a single 2" ram pump system can range from a low of 17 gallons per day to 4,000 gallons per day or more [17].

Apart from the delivery output of the hydraulic ram pump depending on many variables the design itself is complicated by the three pipe flow system and the hydraulic ram effect [18]. The delivery output is a non-linear relationship with variables of input head and output head. Therefore, for a specific hydraulic ram pump, determining the delivery output at variable input and output head heights will be a critical factor in determining the applicability, suitability and effectiveness for use. This study investigates the performance characteristics of a low cost hydraulic ram pump with input and delivery head height variation and quantify the change in efficiency of delivered water.

**Pump Design**
The small hammer-head pump was designed to operate on the hammer head effect created by the sudden stoppage of a flowing fluid. The main components of the pump operation involved two one-way valves and a pressure tank. The one-way valves were arranged such that when one closes the other opened and vice-versa. This design consists of an inlet section followed by the pump body, a pressure section and finally an outlet. A 24.5mm PVC ball valve was installed at the inlet section which allowed control of the water entering the body of the pump and facilitated priming of the pump. The pump was constructed using 32mm diameter PVC pipe and valves. The advantages of this material were low cost, low coefficient of friction and the

resistance to corrosion. Brass one-way swing valves were used in the design. Another 13mm PVC ball valve was placed on
85 the outlet pipe of the pump to prevent back-flow and drainage of the supplied water when the pump was not operating. Figure
1 is a schematic showing the main components of the pump design.

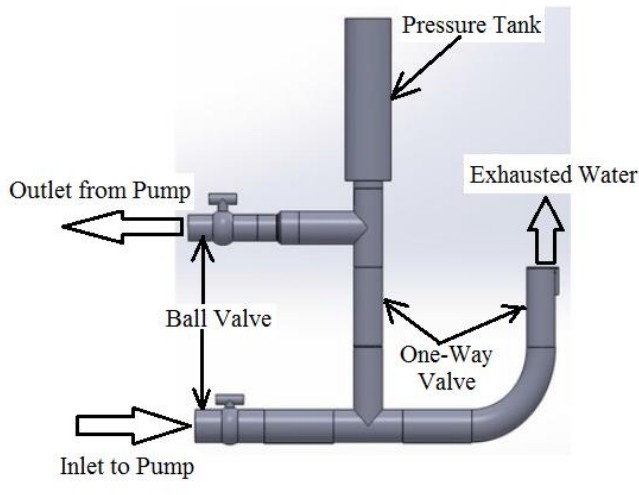 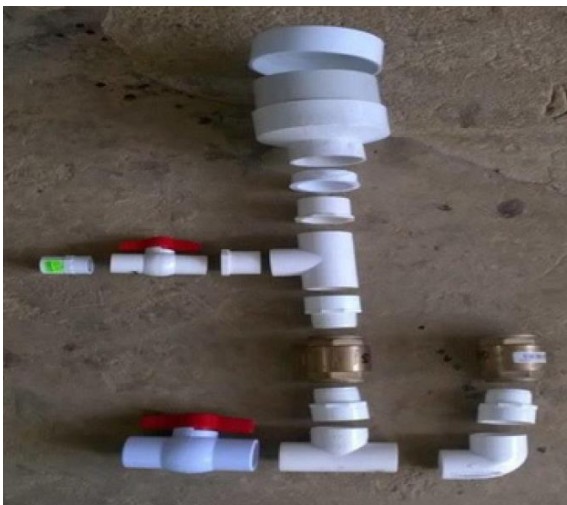

**Fig. 1** Schematic of Hammer Head Pump                **Fig. 2** Picture of pump components

The pressure tank was constructed using a 127mm long 75mm diameter PVC pipe. A PVC end caps was used on one end of
95 the pipe and reduction PVC fittings on the other end attached to the 32mm pipe. Figure 2 is a picture of the pump components
in the position for assembly. The materials/components required for the pump construction were obtained from the local
hardware store. The cost of the components for the pump construction are shown in table 1. The total cost of the pump
components was TT$178, equivalent to US$ 26.

Table 1: Cost of Pump Components

| Component | | TT$ (Trinidad and Tobago dollars) |
|---|---|---|
| 2 | One way swing valve (brass) | 70 |
| 1 | 25.4mm PVC ball valve | 15 |
| 50cm | PVC pipe (32mm diameter) | 5 |
| 1 | 13mm PVC ball valve | 10 |
| 50cm | PVC pipe (75mm diameter) | 10 |
| 2 | PVC end caps (75mm diameter) | 12 |
| 1 | PVC reducer 75mm to 32mm | 8 |
| 1 | PVC reducer 25.4mm to 13mm | 3 |
| 3 | male adapters 32mm | 9 |
| 1 | PVC elbow  32mm | 4 |
| 2 | PVC 'T' 25.4mm/32mm | 20 |
| 1 | PVC male adapter 13mm | 2 |
| 1 | PVC glue 50 ml | 10 |

### Experimental Set-Up

Figure 3 shows a schematic of the experimental apparatus. The experimental set-up for testing the hammer head pump was designed with a variable head input (a) and an adjustable head output (b). The water supply was from a 5000L water reservoir (c). The constant head supply tank was designed with a float (d) that maintained the constant water level as water was supplied to the inlet of the pump. The input head was the difference in height between the inlet of the pump and the water level at the top of the constant head supply tank. The outlet side of the pump used variable length of 13mm diameter PVC pipe to adjust the delivery height (b).

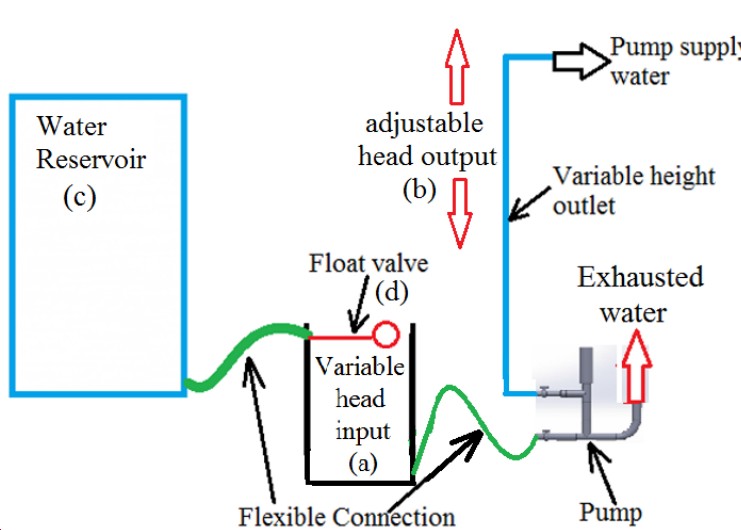

**Fig. 3** Schematic of Experimental Set-up

The water supplied by the pump was collected at fixed time intervals during operation and the volume measured with a 2000ml measuring cylinder with an accuracy of $\pm$20ml to determine the pump supply flow rate. The exhausted water from the pump was also collected at fixed time intervals during operation and the volume measured with a 2000ml measuring cylinder with an accuracy of $\pm$20ml to determine the pump exhausted water flow rate.

### Experimental Results

Experiments were conducted by varying the input head of the water at between 30 cm to 150 cm in increments of 30cm. At each corresponding input head the pump outlet height was varied between 60cm to 600cm in increments of 60cm. For each test configuration, ten samples of pump supply water and pump exhausted water were collected. The water samples were collected for 30s in each case. The volume of water for each sample was measured and the average volume flow rate for the ten samples calculated. This procedure was repeated for each combination of supply head and pump outlet height. The calculated results were tabulated and shown on Tables 2 and 3.

Table 2: Pump output water flow rate variation with input head and outlet height

| Input Head (cm) | Pump Outlet Height (cm) | | | | | | | | | |
|---|---|---|---|---|---|---|---|---|---|---|
| | 60 | 120 | 180 | 240 | 300 | 360 | 420 | 480 | 540 | 600 |
| | Water Flow Rate at Pump Outlet (Pump Supply Water) (ml/min) | | | | | | | | | |
| 30 | 3600 | 2700 | 1200 | 700 | 100 | 0 | 0 | 0 | 0 | 0 |
| 60 | 5600 | 4800 | 3800 | 2700 | 1600 | 1000 | 400 | 200 | 0 | 0 |
| 90 | 7000 | 6800 | 6000 | 4800 | 4200 | 3100 | 2000 | 1500 | 700 | 400 |
| 120 | 8000 | 7800 | 6600 | 5200 | 4400 | 4000 | 3280 | 2400 | 1900 | 1800 |
| 150 | 8800 | 8400 | 8200 | 5600 | 4800 | 4400 | 4000 | 3000 | 2400 | 2200 |

Table 3: Pump exhausted water flow rate variation with input head and outlet height

| Input Head (cm) | Pump Outlet Height (cm) | | | | | | | | | |
|---|---|---|---|---|---|---|---|---|---|---|
| | 60 | 120 | 180 | 240 | 300 | 360 | 420 | 480 | 540 | 600 |
| | Water Flow Rate at Pump Exhaust (Pump Waste Water) (ml/min) | | | | | | | | | |
| 30 | 11600 | 11200 | 16400 | 10600 | 10500 | 0 | 0 | 0 | 0 | 0 |
| 60 | 12800 | 12800 | 15000 | 12600 | 12600 | 12600 | 12400 | 12400 | 12400 | 12400 |
| 90 | 13800 | 13800 | 13600 | 13600 | 13600 | 13400 | 13400 | 13200 | 13200 | 13200 |
| 120 | 14800 | 15200 | 12800 | 15000 | 14800 | 14800 | 14400 | 14400 | 14400 | 14400 |
| 150 | 16000 | 16000 | 11000 | 16400 | 16000 | 15600 | 15200 | 15000 | 14800 | 14400 |

**Analysis and Discussion**

The simple construction low cost water hammer pump showed that as the delivery head increased the rate of water delivered decreased for the five input head height tested. For the lowest input head of 30 cm, the pump operated up to a maximum height of 300 cm. No water was delivered beyond this height. For input head of 60 cm, the pump operated up to a maximum height of 480 cm. No water was delivered beyond this height. For input head of 90 cm, 120 cm and 150 cm, the pump delivered water up to the maximum test height of 600 cm. A plot of the data points indicated a non-linear relationship between pump

outlet height and delivered water flow rate as shown in Figure 4. This observation is in agreement with published literature [16, 18, 19].

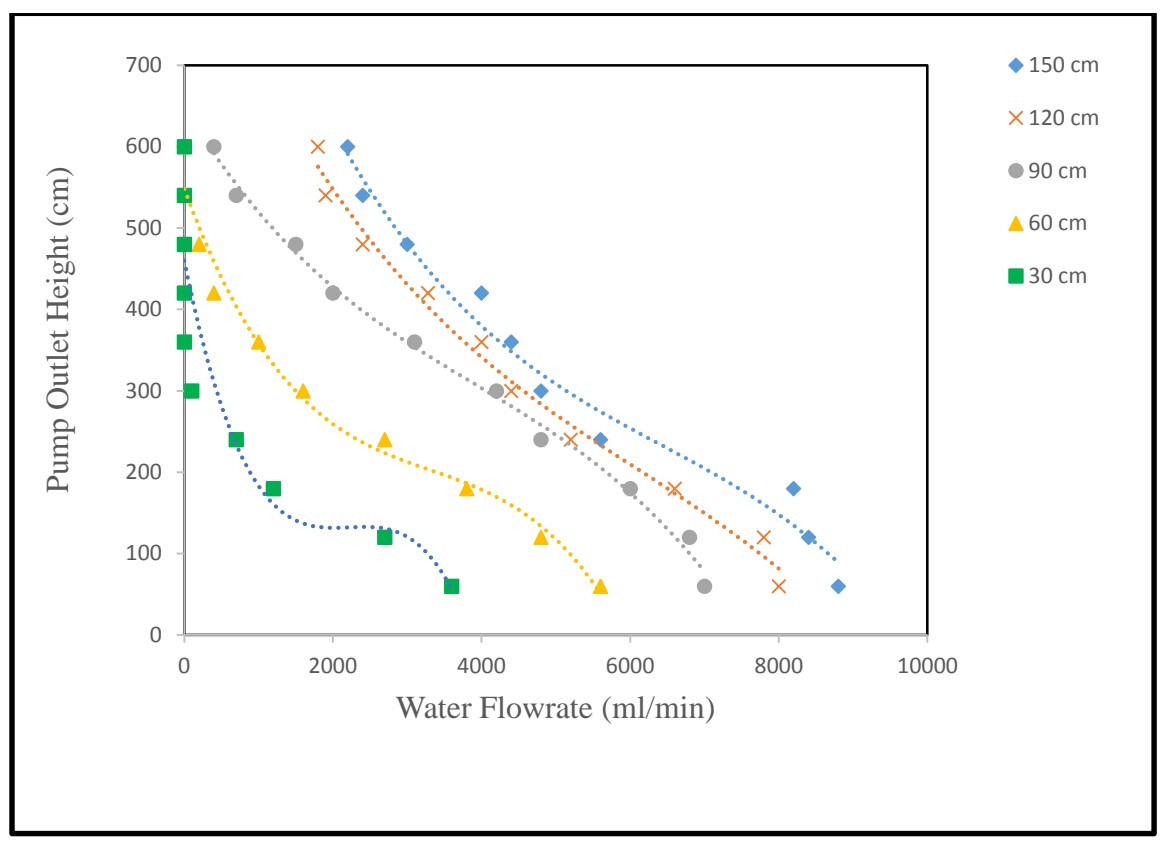

**Fig. 4** Graph of supply water flow rate vs pump outlet height for various input head

The experimental data showed that as pump outlet height decreased, there was a slow increase in delivered water flow rate. This was followed by an increase in delivered water flow rate with a close-to a linear variation. This observation is in agreement with published literature [17, 18]. As the pump outlet height dropped lower than 120 cm, there was a decrease in the delivered water flow rate. The pump efficiency was determined from the ratio of the water delivered to the total water flow. The values were calculated and tabulated on Table 4.

Table 4: Pump efficiency variation with input head and outlet height

| | Pump Outlet Height (cm) | | | | | | | | | |
|---|---|---|---|---|---|---|---|---|---|---|
| | 60 | 120 | 180 | 240 | 300 | 360 | 420 | 480 | 540 | 600 |
| Input Head (cm) | Pump efficiency (%) | | | | | | | | | |
| 30 | 23.7 | 19.4 | 6.8 | 6.2 | 0.9 | 0 | 0 | 0 | 0 | 0 |
| 60 | 30.4 | 27.3 | 20.2 | 17.6 | 11.3 | 7.4 | 13.0 | 1.6 | 0 | 0 |
| 90 | 33.7 | 33.0 | 30.6 | 26.1 | 23.6 | 18.8 | 13.0 | 10.2 | 5.0 | 2.9 |
| 120 | 35.1 | 33.9 | 34.0 | 25.7 | 22.9 | 21.3 | 18.6 | 14.3 | 11.7 | 11.1 |
| 150 | 35.5 | 34.4 | 42.7 | 25.5 | 23.1 | 22.0 | 20.8 | 16.7 | 14.0 | 13.3 |

From the data for the 60 cm and 30 cm input head, the pump was capable of delivering water be between 8 to 10 times the input head with efficiencies of 1.6% and 0.9%, respectively. This was within the range of 5 to 25 times as published by U.S. department of agriculture natural resources conservation service [17]. The difference in delivery height capacity may be due to the shorter pipe length of 300 cm compared to 480 cm associated with the 30 cm and 60 cm input head, respectively. For the increased pipe length there would be increased frictional resistance to flow and the also increased gravitational force due to the higher water column. For input head ranging between 30 cm to 150 cm, the efficiency of the pump delivering water twice the input head height ranged from 23.1% to 30.6% with an average efficiency of 26%. For input head ranging between 30 cm to 150 cm, the efficiency of the pump delivering water four times the input head height ranged from 19.4% to 13.3% with an average efficiency of 16.6%. From the three sets of data available for input head ranging between 30 cm to 90 cm, the efficiency of the pump delivering water six times the input head height ranged from 5.0% to 7.4% with an average efficiency of 6.4%. The trend indicated that as the delivery height increased in increments of twice the corresponding input head, there was a 10% decrease in efficiency. This observation is in conformity with the operation of the hydraulic ram pumps [19].

## Conclusions

- ➢ The delivered water flow rate showed a non-linear variation with respect to pump outlet height.
- ➢ The pump was capable of delivering water to a maximum height of 8 to 10 times the height of the input head.
- ➢ The pump operated at average efficiencies of 26%, 16% and 6% when the delivery height was twice, four times and six times the input head, respectively.
- ➢ There was a 5% incremental decrease in pump efficiency as the delivery height increased in increments of the corresponding input head height.

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
