# Peer review of "Performance Characteristics of a Small Hammer Head Pump"

_Drinking Water Engineering and Science, 2019_

## Referee Comment (RC1) · Anonymous Referee #1 · 20 May 2019

In this paper a design and prototype of a small hammer head pump is presented and the performance of the pump is demonstrated. The topic is of interest and the paper is well written. However, it lacks some (scientific) reasoning, which should be addressed in a next version of the manuscript.

General comments: - A clear objective (and knowledge gap) at the end of the introduction is missing. It should be stated what the drawbacks of the previous designs was, what the research gap is and thus the research question - Use conventional lay-out of manuscript: introduction, Materials and methods, Results and discussion, conclusion - Explain what the innovation in the design of the pump is in the section "pump design and construction" - Relate the results of the experiments to other, similar work (give references) and theory.

[Figure]

Specific comments: - 42, do not use words like "perfect" (and also "very", "a lot" etc.) - 45-46, explain how with "ponds" , "lakes", and "wells" , "a form of flow can be created" - 58, delete "construction" - 60, in Figure 1 the word "exhausted water" and in Figure 3 "waste water" is used. Please synchronize and avoid the word "waste water" since this has another connotation. - 93, explain to what reference the "input head" is related. - 117-127, not of interest, so please delete. - 133-134, explain if this was to be expected (and give reference) - 140-141, explain if this was te be expected (and give reference)

---

## Author Comment (AC1) · 27 May 2019

The comments from reviewer 1 were constructive and serve to improve the quality of the manuscript. The comments are addressed below. These changes will be incorporated in the revised manuscript.

(1) "However, it lacks some (scientific) reasoning, which should be addressed in a next version of the manuscript. General comments: - A clear objective (and knowledge gap) at the end of the introduction is missing. It should be stated what the drawbacks of the previous designs was, what the research gap is and thus the research question"

However, the delivery output of the hydraulic ram pump depends on many variables and is complicated by the three pipe flow system and the hydraulic ram effect [6].

[Figure]

The delivery output is a non-linear relationship with variables of input head and output head. Determining the delivery output of a hydraulic ram pump at variable input and output head heights will be a critical factor in determining the applicability, suitability and effectiveness for use. This study investigates the efficiency of the hydraulic ram pump with input and delivery head height variation and quantify the change in efficiency of delivered water.

[6] Tijsselingc, A. S., and Berganta, A.: Exact computation of water hammer in a three-reservoir system, Eindhoven University of Technology, CASA-Report 12-41:1-10, 2012.

(2) "do not use words like "perfect"" The automatic hydraulic ram has been used for centuries to lift water to heights over 100 meters and is considered an effective machine for 40 pumping water once certain conditions are satisfied.

(3) "explain how with "ponds" , "lakes", and "wells" , "a form of flow can be created"" This source could be a spring, streams, river, ponds, dam, lakes and even some wells, once the conditions exist for these water sources to create a hydraulic flow head.

(4) delete "construction" Brass one-way swing valves were used.

(5) "Figure 1 the word "exhausted water" and in Figure 3 "waste water" is used. Please synchronize and avoid the word "waste water" since this has another connotation." In the revised version; the word waste water will be replaced with exhausted water throughout the manuscript.

(6) "explain to what reference the "input head" is related." The constant head supply tank was designed with a float that maintained the level as water was supplied to the inlet of the pump. The input head was the difference in height between the inlet of the pump and the water level at the top of the constant head supply tank.

(7) "117-127, not of interest, so please delete." The equations will be deleted from the revised manuscript

(8) "explain if this was to be expected (and give reference) - 140-141, explain if this was

to be expected (and give reference)" The trend indicated that as the delivery height increased in increments of twice the corresponding input head, there is a 10% decrease in efficiency. This observation is in conformity with the operation of the hydraulic ram pump [7]. [7] Fatahi-Alkouhi, R., Lashkar-Ara, B., and Keramat, A.: Determine the Efficiency of Hydraulic Ram-Pumps, E-Proceedings of the 36th IAHR World Congress, The Hague, Netherlands, 28th June - 3rd July, 2015.
* * *

---

## Referee Comment (RC2) · Maurits Ertsen (Referee) · 4 Jun 2019

The topic of affordable technology for farmers, that deliver certain services or benefits to them, is of interest. As such, new designs are more then welcome, as long as they are able to show why they are worth being pursued. I am sorry to say that this aspect is missing from the current paper.

Experience with hydro-powered pumps is rather extensive already, and as such one can certainly argue that any new design needs to clarify why its newness brings added value - which could be financial, because it is cheap, or functional, because it delivers water in a specific way, or relate to maintenance, as the pump is easy to repair. None of these aspects are discussed.

[Figure]

The design, the analysis and the numerical statements that are presented remain rather difficult to value when we do not find any information about user prospects, robustness in daily practices, etcetera. I do not want to suggest that the authors need to engage in full-scale field tests first before they can present their own designs. However, just dropping a design with some numbers without explaining why this particular design would be of interest for any target group, is not really appropriate.

Are the different test settings in any way realistic, when we would consider farming practices? Is the discharge in any way useful? What type of use do the authors assume? What additional equipment would a farmer need to make the pump a viable asset on a farm? As long as these question are not at least considered, the information in the paper remains obscure.

A final comment may relate to the number of references. In general, one cannot easily decide what six references mean, but in this case - given the rather high number of documents available on hydro-powered pumps - one would expect a few more references.

---

## Short Comment (SC1) · 4 Jun 2019

General Comments:

Hydrams, since their invention more than two centuries ago, have been attracting many researchers around the world. Therefore it is a relevant topic, but at the same time challenging while pursing for innovation. In this sense, I suggest the authors to highlight as much as possible the actual contribution of their article to this specific field of knowledge. Maybe that contribution is more focused on the easiness of construction and installation, perhaps to its size or maybe to the ratio size / efficiency.

On the other hand, a main part not explicitly addressed in the introduction is the research gap and the consequent research objective. Therefore it is difficult to link the

concluding remarks to the general work.

Moreover, it is highly recommended to keep the traditional structure of a scientific article: Introduction, Materials and methods, Results, Discussion, Conclusion. That aside, it makes more sense to have the "Pump Design and Construction" and "Experimental Set-Up" sections being part of Materials and methods.

Specific comments:

L. 26: Not all the farmlands in the world meet the condition of being far from an electricity source yet close to a water source. Maybe a word like "usually" or "mostly" could fit, as long as evidence is provided for such an affirmation.

L. 26: It is better to explain why the relevance of being far from electricity sources, i.e. why is it a challenge/problem for farming. Besides, please think of what happens with diesel-based pumps, which do not rely on electricity.

L. 27: Is there evidence for stating "The water source is usually situated below the level of the farmlands"? Or perhaps it is better to say something more general, such as "there are cases where the water source...".

L. 28: There is the need of an introductory / transition sentence before "A water pump operating on the water hammer effect...". I suggest to introduce the reader why it is a challenge being far from electricity, and what can be done using hydropower. Then the explanation of the hydraulic ram pump will fit better in the storyline.

Ls. 31-38: The historic introduction, particularly if it does not go beyond the work done by Pierre Mongolfier, must be briefly summarized. Its constructive details are not relevant for the scope of the paper.

L. 39: In principle, no machine can be considered "perfect". Furthermore, what are the criteria to be considered as such? I recommend to use expressions like "highly reliable", or any other that reflects its degree of development. In adittion, hydrams, compared to other similar technologies, are subject to constant wearing due to the

aggressiveness of the water hammer effect, which is in turn their main drawback.

L. 41: It mentions "water was wasted", whereas the Fig. 1 refers to "exhausted water". It is important to keep consistency in the nomenclature, and making sure it matches with the usual terminology in the literature (e.g. "A Manual On The Hydraulic Ram For Pumping Water" by S.B. Watt, or "Hydraulic Ram Pumps: A guide to ram pump water supply systems" by Jeffrey et al.)

L. 43: About "once any form of flow can be created", it will be good to provide an insight on how this flow can be created after the different water sources, i.e. what kind of extra infrastructure it might need: dam, weir, drop, etc.

L. 44: The ram pump installed at a lower location than the water source is not an ideal scenario but a mandatory one. Otherwise the pump will not operate whatsoever.

L. 48-41: This paragraph describes the generic structure of a Hydram. I suggest to put that in the introduction, so in this section the specific parts and assembly methods of your prototype are directly described.

L. 61: Figs 1 and 2 could be put side to side, so the reader can make a quick correspondence between the scheme and the actual prototype.

L. 73-76: I recommend to match the parts of the experimental set-up, as described in this paragraph, with those of the Fig. 3, to make sure all of them can be identified in both graphic and text. A good way of achieving it could be by assigning letters or numbers to each part.

L. 93: I wonder if it would be more convenient to combine both tables 1 and 2 in a single one, due to their similar structure. In that case, each cell must be divided in two parts, for the pumped flow and wasted flow, respectively. Moreover, this can give the chance to include a third part: the pumped/wasted ratio; it can be eventually related to the pumping efficiency.

L. 100 and next ones: The discussion part must be enriched by comparing your study

with other ones, perhaps using similar prototypes in sizes and conditions. Of course, the respective literature and references must be taken into account while doing so.

L. 134: The first conclusion might be too obvious, after so many years of continuous and ubiquitous use of the hydrams, to be considered as such after the present study.

Technical corrections:

Keywords: "Pump characteristic" is not a so accurate keyword for this study, since it does not suggest any possible topic.

L. 26: Farmlands are (missing plural).

L. 37: The name of the son of Joseph Michel de Montgolfier, who improved his father's model, is Pierre (or Pierre François).

L. 37: "Montgolfier designed the sniffer valve that reintroduce..." It must be "reintroduced".

L. 40: "The pump construction was simple and consisted of a pump camber...". Do you refer to "chamber" perhaps?

---

## Author Comment (AC2) · 2 Aug 2019

The introduction section of the paper was enhanced to address the constructive comments received from reviewer 2. More specific: Referee comments: "Experience with hydro-powered pumps is rather extensive already, and as such one can certainly argue that any new design needs to clarify why its newness brings added value - which could be financial, because it is cheap, or functional, because it delivers water in a specific way, or relate to maintenance, as the pump is easy to repair. None of these aspects are discussed."

Author's response:'The major hindrance in using this established technology in third world countries is the exorbitant cost of the commercially available units. For a UK

built pump the cost is US$ 1800 [10] and cheaper china made pumps range between US$500 to US$1300 [11]. One of the objectives of the Prude University project in Haiti was to develop a cheaper alternative, however, the cost was US$100 [9]. Therefore, there is the need to develop a low cost alternative that can be easily built from readily construction materials and requires minimal technical skills.'

Referee comments: "The design, the analysis and the numerical statements that are presented remain rather difficult to value when we do not find any information about user prospects, robustness in daily practices, etcetera. I do not want to suggest that the authors need to engage in full-scale field tests first before they can present their own designs. However, just dropping a design with some numbers without explaining why this particular design would be of interest for any target group, is not really appropriate."

Author's response:'In many rural farming areas, having a reliable source of water for crops and livestock can prove to be an expensive venture. In developing and under-developed countries, farmland are usually located close to a reliable water source to ensure viability [6, 7]. However, in many instances these locations are far from any reliable source of electricity and the cost can be prohibitive [6, 8]. In cases where the water source is situated below the level of the farmlands, getting the water to where it is needed can be challenging. Under these circumstances, a water pump operating on the water hammer effect and requires no external power source can serve as an effective means of pumping water to a higher altitude, once a reliable source is available. Also, in under developed countries, such as Haiti, the feasibility of using small hammer head pumps to provide clean water for citizens were explored by Prude University [9].'

Referee comments:"Are the different test settings in any way realistic, when we would consider farming practices? "

Authors response: 'The delivery output is a non-linear relationship with variables of input head and output head. Therefore, for a specific hydraulic ram pump, determining the delivery output at variable input and output head heights will be a critical factor in

determining the applicability, suitability and effectiveness for use.'

Referee comments: "Is the discharge in any way useful? What type of use do the authors assume? What additional equipment would a farmer need to make the pump a viable asset on a farm?"

Authors response: 'Also, in under developed countries, such as Haiti, the feasibility of using small hammer head pumps to provide clean water for citizens were explored by Prude University [9]. The ram pump can operate 24/7 and hence a water storage facility, such as storage tanks, at the water delivery end will be needed. This will serve as the reservoir to supply the needs when required.'

Referee comments: "A final comment may relate to the number of references. In general, one cannot easily decide what six references mean, but in this case - given the rather high number of documents available on hydro-powered pumps - one would expect a few more references."

Authors response: 'The number of references were increased to sixteen as shown in the revised introduction section below.'

[revised manuscript text omitted]
 Investigators: Engel, Bernard A. , Ahiablame, Laurent , DeNardo, Nick , Deak, Brian , Garner, Leah, Kujur, Birendra , Poppe, Brooke Institution: Purdue University EPA Project Officer: Hahn, Intaek Project Period: August 15, 2011 through August 14, 2012 [10] 2019, Papa Hydraulic Ram Pump, Water Powered, Fuelless, Gravity, Eco Pump Kit - 2 inch including Seradisc Filters. (available at https://www.amazon.com/Papa-Pump-Hydraulic-Water-system/dp/B06XXTMF4Q) [11] 2019, 24 hours uninterrupted lift water machine automatic hydraulic ram pump. (available at https://www.alibaba.com/product-detail/24-hours-uninterrupted-lift-watermachine_60572542323.html?spm=a2700.7724857.normalList.2.20173dbdfO4fjp
[12] Advances in Civil Engineering Volume 2019, Article ID 9702183, https://doi.org/10.1155/2019/9702183 Determination of Hydraulic Ram Pump Performance: Experimental Results Wanchai Asvapoositkul, Jedsada Juruta, Nattapong Tabtimhin, and Yosawat Limpongsa [13] B. W. Young, "Simplified analysis and design of the hydraulic ram pump," in Proceedings of the Institution of Mechanical Engineers, Part A: Journal of Power and Energy, vol. 210, no. 4, pp. 295–303, 2016. [14] W. M. Lansford and W. G. Dugan, "An analytical and experimental study of the hydraulic ram," University of Illinois Bulletin, vol. 38, no. 22, pp. 1–70, 1941. [15] TECHNICAL NOTES: U.S. DEPARTMENT OF AGRICULTURE NATURAL RESOURCES CONSERVATION SERVICE PORTLAND, OREGON, SEPTEMBER 2007, RANGE TECHNICAL NOTE NO. 26 Hydraulic Ram Pumps (available at https://www.nrcs.usda.gov/Internet/FSE_DOCUMENTS/nrcs142p2_041913.pdf) [16] Tijsselingc, A. S., and Berganta, A.: Exact computation of water hammer in a three reservoir system, Eindhoven University of Technology, CASA-Report 12-41:1-10, 2012.

This cost list was included The materials/components required for the pump construction were obtained from the local hardware store. The cost of the components for the pump construction are shown in table 1. The total cost of the pump components is TT$178, which is equivalent to US$ 26.

Component TT$ (Trinidad and Tobago dollars) 2 One way swing valve (brass) 70 1 25.4mm PVC ball valve 15 50cm PVC pipe (32mm diameter) 5 1 13mm PVC ball valve 10 50cm PVC pipe (75mm diameter) 10 2 PVC end caps (75mm diameter) 12 1 PVC reducer 75mm to 32mm 8 1 PVC reducer 25.4mm to 13mm 3 3 male adapters 32mm 9 1 PVC elbow 32mm 4 2 PVC 'T' 25.4mm/32mm 20 1 PVC male adapter 13mm 2 1 PVC glue 50 ml 10

2019.

---

## Author Comment (AC3) · 2 Aug 2019

Reviewer comments: "In this sense, I suggest the authors to highlight as much as possible the actual contribution of their article to this specific field of knowledge. Maybe that contribution is more focused on the easiness of construction and installation, perhaps to its size or maybe to the ratio size / efficiency. On the other hand, a main part not explicitly addressed in the introduction is the research gap and the consequent research objective. Therefore it is difficult to link the concluding remarks to the general work."

Authors response: "The major hindrance in using this established technology in third world countries is the exorbitant cost of the commercially available units. For a UK

built pump the cost is US$ 1800 [10] and cheaper china made pumps range between US$500 to US$1300 [11]. One of the objectives of the Prude University project in Haiti was to develop a cheaper alternative, however, the cost was US$100 [9]. Therefore, there is the need to develop a low cost alternative that can be easily built from readily construction materials and requires minimal technical skills."

Reviewer comments: " Not all the farmlands in the world meet the condition of being far from an electricity source yet close to a water source. Maybe a word like "usually" or "mostly" could fit, as long as evidence is provided for such an affirmation."

Authors response: "In developing and under-developed countries, farmland are usually located close to a reliable water source to ensure viability [6, 7]."

Reviewer comments: "It is better to explain why the relevance of being far from electricity sources, i.e. why is it a challenge/problem for farming. Besides, please think of what happens with diesel-based pumps, which do not rely on electricity."

Authors response: "However, in many instances these locations are far from any reliable source of electricity and the cost can be prohibitive [6, 8]."

Reviewers comments: " Is there evidence for stating "The water source is usually situated below the level of the farmlands"? Or perhaps it is better to say something more general, such as "there are cases where the water source..."."

Authors response: "In cases where the water source is situated below the level of the farmlands, getting the water to where it is needed can be challenging [7]. "

Reviewers comments: There is the need of an introductory / transition sentence before "A water pump operating on the water hammer effect...". I suggest to introduce the reader why it is a challenge being far from electricity, and what can be done using hydropower. Then the explanation of the hydraulic ram pump will fit better in the storyline."

Authors response: "In cases where the water source is situated below the level of the farmlands, getting the water to where it is needed can be challenging [7]. Under these circumstances, a water pump operating on the water hammer effect and requires no external power source can serve as an effective means of pumping water to a higher altitude, once a reliable source is available."

Reviewers comments: " The historic introduction, particularly if it does not go beyond the work done by Pierre Mongolfier, must be briefly summarized. Its constructive details are not relevant for the scope of the paper."

Authors response: "Given the long history of the hydraulic ram pump, the design and manufacture has improved considerably with time and efficiency of operation increased. For commercial ram pumps the typical energy efficiency is about 60%, but can reach up to 80% [12].This is different from the volumetric efficiency, which relates the volume of water delivered to total water taken from the source. The amount of water delivered will be reduced by the ratio of the output head to the supply head. For example, if the source is 2 meters above the ram pump and the water is lifted to 10 meters above, only 20% of the supplied water will be available and the other 80% being spilled via the waste valve [13]."

Reviewers comments: " In principle, no machine can be considered "perfect". Furthermore, what are the criteria to be considered as such? I recommend to use expressions like "highly reliable", or any other that reflects its degree of development. In adittion, hydrams, compared to other similar technologies, are subject to constant wearing due to the aggressiveness of the water hammer effect, which is in turn their main drawback."

Authors response: "The automatic hydraulic ram has been used for centuries to lift water to heights over 100 meters and is considered an effective machine for pumping water once certain conditions are satisfied."

Reviewers comments: " It mentions "water was wasted", whereas the Fig. 1 refers to "exhausted water". It is important to keep consistency in the nomenclature, and making sure it matches with the usual terminology in the literature (e.g. "A Manual On

The Hydraulic Ram For Pumping Water" by S.B. Watt, or "Hydraulic Ram Pumps: A guide to ram pump water supply systems" by Jeffrey et al.)"

Authors response: "This editorial change will be made to ensure consistency in the manuscript."

Reviewers comments: ": About "once any form of flow can be created", it will be good to provide an insight on how this flow can be created after the different water sources, i.e. what kind of extra infrastructure it might need: dam, weir, drop, etc."

Authors response: "This source could be a spring, streams, river, ponds, dam, lakes and even some wells, once the conditions exist for these water sources to create a hydraulic flow head, either by forming a dam or a naturally existing head."

Reviewers comments: "The ram pump installed at a lower location than the water source is not an ideal scenario but a mandatory one. Otherwise the pump will not operate whatsoever."

Authors response: "The ram pump must be installed at a location lower than the water source which is used to create the flow giving the fluid (water) some velocity."

Reviewers Comments: This paragraph describes the generic structure of a Hydram. I suggest to put that in the introduction, so in this section the specific parts and assembly methods of your prototype are directly described.

L. 61: Figs 1 and 2 could be put side to side, so the reader can make a quick correspondence between the scheme and the actual prototype.

L. 73-76: I recommend to match the parts of the experimental set-up, as described in this paragraph, with those of the Fig. 3, to make sure all of them can be identified in both graphic and text. A good way of achieving it could be by assigning letters or numbers to each part.

L. 93: I wonder if it would be more convenient to combine both tables 1 and 2 in a single one, due to their similar structure. In that case, each cell must be divided in two parts, for the pumped flow and wasted flow, respectively. Moreover, this can give the chance to include a third part: the pumped/wasted ratio; it can be eventually related to the pumping efficiency.

L. 100 and next ones: The discussion part must be enriched by comparing your study with other ones, perhaps using similar prototypes in sizes and conditions. Of course, the respective literature and references must be taken into account while doing so.

L. 134: The first conclusion might be too obvious, after so many years of continuous and ubiquitous use of the hydrams, to be considered as such after the present study.

Technical corrections: Keywords: "Pump characteristic" is not a so accurate keyword for this study, since it does not suggest any possible topic. L. 26: Farmlands are (missing plural). L. 37: The name of the son of Joseph Michel de Montgolfier, who improved his father's model, is Pierre (or Pierre François). L. 37: "Montgolfier designed the sniffer valve that reintroduce..." It must be "reintroduced". L. 40: "The pump construction was simple and consisted of a pump camber...". Do you refer to "chamber" perhaps?

Authors response: All the editorial and technical corrections will be done in the final manuscript.

Please also note the supplement to this comment:
https://www.drink-water-eng-sci-discuss.net/dwes-2019-7/dwes-2019-7-AC3-supplement.pdf

**Supplement:**

[revised manuscript text omitted]

        Investigators: Engel, Bernard A. , Ahiablame, Laurent , DeNardo, Nick , Deak, Brian , Garner, Leah, Kujur, Birendra , Poppe, Brooke
        Institution: Purdue University
        EPA Project Officer: Hahn, Intaek
        Project Period: August 15, 2011 through August 14, 2012

[10]   2019, Papa Hydraulic Ram Pump, Water Powered, Fuelless, Gravity, Eco Pump Kit - 2 inch
including Seradisc Filters. (available at https://www.amazon.com/Papa-Pump-Hydraulic-Water-
system/dp/B06XXTMF4Q)
[11]   2019, 24 hours uninterrupted lift water machine automatic hydraulic ram pump. (available at
https://www.alibaba.com/product-detail/24-hours-uninterrupted-lift-water-
machine_60572542323.html?spm=a2700.7724857.normalList.2.20173dbdfO4fjp

[12]   Advances in Civil Engineering
Volume 2019, Article ID 9702183,
https://doi.org/10.1155/2019/9702183
Determination of Hydraulic Ram Pump Performance: Experimental Results
Wanchai Asvapoositkul, Jedsada Juruta, Nattapong Tabtimhin, and Yosawat Limpongsa

[13]   B. W. Young, "Simplified analysis and design of the hydraulic ram pump," in Proceedings of the
Institution of Mechanical Engineers, Part A: Journal of Power and Energy, vol. 210, no. 4, pp. 295–
303, 2016.

[14]   W. M. Lansford and W. G. Dugan, "An analytical and experimental study of the hydraulic
ram," University of Illinois Bulletin, vol. 38, no. 22, pp. 1–70, 1941.

[15]   TECHNICAL NOTES: U.S. DEPARTMENT OF AGRICULTURE NATURAL RESOURCES
CONSERVATION SERVICE PORTLAND, OREGON, SEPTEMBER 2007, RANGE
TECHNICAL NOTE NO. 26 Hydraulic Ram Pumps (available at
https://www.nrcs.usda.gov/Internet/FSE_DOCUMENTS/nrcs142p2_041913.pdf)

[16]   Tijsselingc, A. S., and Berganta, A.: Exact computation of water hammer in a three reservoir system,
Eindhoven University of Technology, CASA-Report 12-41:1-10, 2012.

*This cost list was included*

The materials/components required for the pump construction were obtained from the local hardware store.  The cost of the
components for the pump construction are shown in table 1.  The total cost of the pump components is TT$178, which is equivalent
to US$ 26.

| Component | | TT$ (Trinidad and Tobago dollars) |
|---|---|---|
| 2 | One way swing valve (brass) | 70 |
| 1 | 25.4mm PVC ball valve | 15 |
| 50cm | PVC pipe (32mm diameter) | 5 |
| 1 | 13mm PVC ball valve | 10 |
| 50cm | PVC pipe (75mm diameter) | 10 |
| 2 | PVC end caps (75mm diameter) | 12 |
| 1 | PVC reducer 75mm to 32mm | 8 |
| 1 | PVC reducer 25.4mm to 13mm | 3 |
| 3 | male adapters 32mm | 9 |
| 1 | PVC elbow  32mm | 4 |
| 2 | PVC 'T' 25.4mm/32mm | 20 |
| 1 | PVC male adapter 13mm | 2 |
| 1 | PVC glue 50 ml | 10 |

---

## Author Response (AR1)

[revised manuscript text omitted]

*The comments from reviewer 1 were constructive and served to improve the quality of the manuscript. The comments are addressed below.*

*Referee comment: However, it lacks some (scientific) reasoning, which should be addressed in a next version of the manuscript. General comments: - A clear objective (and knowledge gap) at the end of the introduction is missing. It should be stated what the drawbacks of the previous designs was, what the research gap is and thus the research question*

Author response: (marked up manuscript lines 95-101)

Apart from the delivery output of the hydraulic ram pump depending on many variables the design itself is complicated by the three pipe flow system and the hydraulic ram effect [17]. The delivery output is a non-linear relationship with variables of input head and output head. Therefore, for a specific hydraulic ram pump, determining the delivery output at variable input and output head heights will be a critical factor in determining the applicability, suitability and effectiveness for use. This study investigates the performance characteristics of a low cost hydraulic ram pump with input and delivery head height variation and quantify the change in efficiency of delivered water.

[17]  Tijsselingc, A. S., and Berganta, A.: Exact computation of water hammer in a three-reservoir system, Eindhoven University of Technology, CASA-Report 12-41:1-10, 2012.

*Referee comment: A clear objective (and knowledge gap) at the end of the introduction is missing. It should be stated what the drawbacks of the previous designs was, what the research gap is and thus the research question –*

Author response: (marked up manuscript lines 54-63)

Also, in under developed countries, such as Haiti, the feasibility of using small hammer head pumps to provide clean water for citizens were explored by Prude University [9]. The ram pump can operate 24/7 and hence a water storage facility, such as storage tanks, at the water delivery end will be needed. This will serve as the reservoir to supply the needs when required. The major hindrance in using this established technology in third world countries is the exorbitant cost of the commercially available units. For a UK built pump the cost is US$ 1800 [10] and cheaper china made pumps range between US$500 to US$1300 [11]. One of the objectives of the Prude University project in Haiti was to develop a cheaper alternative, however, the cost was US$100 [9]. Therefore, there is the need to develop a low cost

alternative that can be easily built from readily available construction materials and requires minimal
technical skills.

Also, above - marked up manuscript lines 95-101

***Referee comment:*** *do not use words like "perfect"*

Author response: (marked up manuscript lines 33-35)

The automatic hydraulic ram has been used for centuries to lift water to heights over 100 meters and was considered an effective and highly reliable machine for pumping water once certain conditions are satisfied.

***Referee comment:*** *explain how with "ponds", "lakes", and "wells" , "a form of flow can be created"*

Author response: (marked up manuscript lines 38-41)

This source could be a spring, streams, river, ponds, dam, lakes and even some wells, once the conditions existed for these water sources to create a hydraulic flow head, either by forming a dam or a naturally existing head. Basically, once a hydraulic head can be created, the pump can operate, however, the source must provide a steady and reliable supply of water [5].

***Referee comment:*** *delete "construction"*

Author response: (marked up manuscript line 110)

Brass one-way swing valves were used in the design

***Referee comment:*** *Figure 1 the word "exhausted water" and in Figure 3 "waste water" is used. Please synchronize and avoid the word "waste water" since this has another connotation. –*

Author response:

In the marked up manuscript version; the word waste water was replaced with exhausted water throughout the manuscript.

***Referee comment:*** *explain to what reference the "input head" is related.*

Author response: (marked up manuscript lines 139-141)

The constant head supply tank was designed with a float (d) that maintained the constant water level as water was supplied to the inlet of the pump. The input head was the difference in height between the inlet of the pump and the water level at the top of the constant head supply tank.

***Referee comment:*** *117-127, not of interest, so please delete.*

Author response: (marked up manuscript lines 178-189)

The equations were deleted from the revised marked up manuscript

***Referee comment:*** *"explain if this was to be expected (and give reference) - 140-141,*

Author response: (marked up manuscript lines 194-196)

From the data for the 60 cm and 30 cm input head, the pump was capable of delivering water be between

8 to 10 times the input head with efficiencies of 1.6% and 0.9%, respectively. This was within the range of 5 to 25 times as published by U.S. department of agriculture natural resources conservation service [16].

***Referee comment:*** *explain if this was to be expected (and give reference)"*

Author response: (marked up manuscript lines 204-206)

The trend indicated that as the delivery height increased in increments of twice the corresponding input head, there is was a 10% decrease in efficiency. This observation is in conformity with the operation of the hydraulic ram pumps [18].

*The comments from reviewer 2 were constructive and served to improve the quality of the manuscript. The comments are addressed below.*

**_Referee comment:_** *The topic of affordable technology for farmers, that deliver certain services or benefits to them, is of interest. As such, new designs are more then welcome, as long as they are able to show why they are worth being pursued. I am sorry to say that this aspect is missing from the current paper. Experience with hydro-powered pumps is rather extensive already, and as such one can certainly argue that any new design needs to clarify why its newness brings added value - which could be financial, because it is cheap, or functional, because it delivers water in a specific way, or relate to maintenance, as the pump is easy to repair. None of these aspects are discussed.*

Author response: (marked up manuscript lines 54-63)

Also, in under developed countries, such as Haiti, the feasibility of using small hammer head pumps to provide clean water for citizens were explored by Prude University [9]. The ram pump can operate 24/7 and hence a water storage facility, such as storage tanks, at the water delivery end will be needed.  This will serve as the reservoir to supply the needs when required. The major hindrance in using this established technology in third world countries is the exorbitant cost of the commercially available units. For a UK built pump the cost is US$ 1800 [10] and cheaper china made pumps range between US$500 to US$1300 [11].  One of the objectives of the Prude University project in Haiti was to develop a cheaper alternative, however, the cost was US$100 [9].  ==Therefore, there is the need to develop a low cost alternative that can be easily built from readily available construction materials and requires minimal technical skills.==

(marked up manuscript lines 108-110)

The pump was constructed using 32mm diameter PVC pipe and valves.  The advantages of this material were  low cost, low coefficient of friction and the resistance to corrosion.  Brass one-way swing valves were used in the design.

(marked up manuscript lines 95-101)

Apart from the delivery output of the hydraulic ram pump depending on many variables the design itself is complicated by the three pipe flow system and the hydraulic ram effect [17]. The delivery output is a non-linear relationship with variables of input head and output head. Therefore, for a specific hydraulic ram pump, determining the delivery output at variable input and output head heights will be a critical factor in determining the applicability, suitability and effectiveness for use. This study investigates the

performance characteristics of a low cost hydraulic ram pump with input and delivery head height variation and quantify the change in efficiency of delivered water.

*Referee comment: The design, the analysis and the numerical statements that are presented remain rather difficult to value when we do not find any information about user prospects, robustness in daily practices, etcetera. I do not want to suggest that the authors need to engage in full-scale field tests first before they can present their own designs. However, just dropping a design with some numbers without explaining why this particular design would be of interest for any target group, is not really appropriate.*

Author response: above section (marked up manuscript lines 54-63)

*Referee comment: Are the different test settings in any way realistic, when we would consider farming practices? Is the discharge in any way useful? What type of use do the authors assume? What additional equipment would a farmer need to make the pump a viable asset on a farm? As long as these question are not at least considered, the information in the paper remains obscure.*

Author response: (marked up manuscript lines 54-58)

Also, in under developed countries, such as Haiti, the feasibility of using small hammer head pumps to provide clean water for citizens were explored by Prude University [9]. The ram pump can operate 24/7 and hence a water storage facility, such as storage tanks, at the water delivery end will be needed. This will serve as the reservoir to supply the needs when required.

(marked up manuscript lines 96-101)

The delivery output is a non-linear relationship with variables of input head and output head. Therefore, for a specific hydraulic ram pump, determining the delivery output at variable input and output head heights will be a critical factor in determining the applicability, suitability and effectiveness for use. This study investigates the performance characteristics of a low cost hydraulic ram pump with input and delivery head height variation and quantify the change in efficiency of delivered water.

*Referee comment: A final comment may relate to the number of references. In general, one cannot easily decide what six references mean, but in this case - given the rather high number of documents available on hydro-powered pumps - one would expect a few more references.*

Author response: (marked up manuscript lines 218-270)

The number of references were increased to 1

*The comments from reviewer 3 were constructive and served to improve the quality of the manuscript. The comments are addressed below.*

***Referee comment:*** *General Comments: Hydrams, since their invention more than two centuries ago, have been attracting many researchers around the world. Therefore it is a relevant topic, but at the same time challenging while pursing for innovation. In this sense, I suggest the authors to highlight as much as possible the actual contribution of their article to this specific field of knowledge. Maybe that contribution is more focused on the easiness of construction and installation, perhaps to its size or maybe to the ratio size / efficiency.*

Author response: (marked up manuscript lines 58-63)

The major hindrance in using this established technology in third world countries is the exorbitant cost of the commercially available units. For a UK built pump the cost is US$ 1800 [10] and cheaper china made pumps range between US$500 to US$1300 [11]. One of the objectives of the Prude University project in Haiti was to develop a cheaper alternative, however, the cost was US$100 [9]. Therefore, there is the need to develop a low cost alternative that can be easily built from readily available construction materials and requires minimal technical skills.

(marked up manuscript lines 128-133)

The materials/components required for the pump construction were obtained from the local hardware store. The cost of the components for the pump construction are shown in table 1. The total cost of the pump components was TT$178, equivalent to US$ 26.

| Component | | TT$ (Trinidad and Tobago dollars) |
|---|---|---|
| 2 | One way swing valve (brass) | 70 |
| 1 | 25.4mm PVC ball valve | 15 |
| 50cm | PVC pipe (32mm diameter) | 5 |
| 1 | 13mm PVC ball valve | 10 |
| 50cm | PVC pipe (75mm diameter) | 10 |
| 2 | PVC end caps (75mm diameter) | 12 |

| 1 | PVC reducer 75mm to 32mm | 8 |
|---|---|---|
| 1 | PVC reducer 25.4mm to 13mm | 3 |
| 3 | male adapters 32mm | 9 |
| 1 | PVC elbow  32mm | 4 |
| 2 | PVC 'T' 25.4mm/32mm | 20 |
| 1 | PVC male adapter 13mm | 2 |
| 1 | PVC glue 50 ml | 10 |

***Referee comment:*** *On the other hand, a main part not explicitly addressed in the introduction is the research gap and the consequent research objective. Therefore it is difficult to link the C1 DWESD Interactive comment Printer-friendly version Discussion paper concluding remarks to the general work.*

Author response: (marked up manuscript lines 46-59)

In developing and under-developed countries, farmlands are ideally located close to a reliable water source to ensure viability [6, 7]. However, in many instances these locations are far from any reliable source of electricity and the cost can be prohibitive [6, 8].  The the water source is  situated below the level of the farmlands,  getting the water to where it is needed can be challenging [7].  Under these circumstances,  a water pump operating on the water hammer effect and requires no external power source  can serve as an effective means of pumping water to a higher altitude once a reliable source is available. Also, in under developed countries, such as Haiti, the feasibility of using small hammer head pumps to provide clean water for citizens were explored by Prude University [9]. The ram pump can operate 24/7 and hence a water storage facility, such as storage tanks, at the water delivery end will be needed.  This will serve as the reservoir to supply the needs when required. The major hindrance in using this established technology in third world countries is the exorbitant cost of the commercially available units.

***Referee comment:*** *Moreover, it is highly recommended to keep the traditional structure of a scientific article: Introduction, Materials and methods, Results, Discussion, Conclusion. That aside, it makes more sense to have the "Pump Design and Construction" and "Experimental Set-Up" sections being part of Materials and methods.*

Author response: In my opinion, I would prefer to leave the manuscript in the format presented, unless it is a mandatory requirement of the journal format.

480 ***Referee comment:*** *L. 26: Not all the farmlands in the world meet the condition of being far from an electricity source yet close to a water source. Maybe a word like "usually" or "mostly" could fit, as long as evidence is provided for such an affirmation.*

Author response: (marked up manuscript lines 44-51)

In many rural farming areas, having a reliable source of water for crops and livestock can prove to be an
485 expensive venture.  In developing and under-developed countries, farmlands are ideally located close to a reliable water source to ensure viability [6, 7]. However, in many instances these locations are far from any reliable source of electricity and the cost can be prohibitive [6, 8]. The alternative of diesel-driven pumps create high operation costs and are prone to service gaps due
490 to insufficient fuel supply and technical defects. A reliable and cost-effective supply of irrigation water is therefore a core problem in many rural areas in developing and emerging countries [9].

***Referee comment:*** *L. 26: It is better to explain why the relevance of being far from electricity sources, i.e. why is it a challenge/problem for farming. Besides, please think of what happens with diesel-based pumps, which do not rely on electricity.*

495 Author response: (marked up manuscript lines 44-51) above

***Referee comment:*** *L. 27: Is there evidence for stating "The water source is usually situated below the level of the farmlands"? Or perhaps it is better to say something more general, such as "there are cases where the water source...".*

Author response: (marked up manuscript lines 51-52)

500 In cases where  the water source is  situated below the level of the farmlands,  getting the water to where it is needed can be challenging [7].

***Referee comment:*** *L. 28: There is the need of an introductory / transition sentence before "A water pump operating on the water hammer effect...". I suggest to introduce the reader why it is a challenge being far
505 from electricity, and what can be done using hydropower. Then the explanation of the hydraulic ram pump will fit better in the storyline.*

In many rural farming areas, having a reliable source of water for crops and livestock can prove to be an expensive venture.   In developing and under-developed countries, farmlands are ideally located close to a reliable water source to ensure viability [6, 7]. However, in many instances these locations are far from any reliable source of electricity and the cost can be prohibitive [6, 8]. The alternative of diesel-driven pumps create high operation costs and are prone to service gaps due to insufficient fuel supply and technical defects. A reliable and cost-effective supply of irrigation water is therefore a core problem in many rural areas in developing and emerging countries [9]. In cases where  the water source is  situated below the level of the farmlands,  getting the water to where it is needed can be challenging [7].  Under these circumstances,  a water pump operating on the water hammer effect and requires no external power source  can serve as an effective means of pumping water to a higher altitude once a reliable source is available.

*Referee comment:*  *Ls. 31-38: The historic introduction, particularly if it does not go beyond the work done by Pierre Mongolfier, must be briefly summarized. Its constructive details are not relevant for the scope of the paper.*

The first type of pumps to use the water hammer effect was the hydraulic ram pump which was reported in 1775 and was built by John Whitehurst [1]. His design was not automatic and was controlled by manually opening and closing a stopcock which resulted in the device only being able to raise water to a height of 4.9 meters. This involved a significant amount of work and consumed a lot of time to operate. However, in 1797 the design was improved and the first reported automatic hydraulic ram was developed by Joseph and Etienne Montgolfier to raise water to a paper mill [2]. Although this was an improved design it still contained design flaws which caused the air in the pressure chamber to dissolve or drop. In 1816 this problem was eliminated when Pierre Montgolfier designed the sniffer valve that reintroduced air into the chamber. This valve was 15 cm in radius and it was reported that the pump was able to raise water to 48 meters in height [3]. The automatic hydraulic ram has been used for centuries to lift water to heights over 100 meters and was considered an effective and highly reliable machine for pumping water once certain conditions are satisfied.

*Referee comment:*  *L. 39: In principle, no machine can be considered "perfect". Furthermore, what are the criteria to be considered as such? I recommend to use expressions like "highly reliable", or any other that reflects its degree of development. In adittion, hydrams, compared to other similar technologies, are*

540 *subject to constant wearing due to the aggressiveness of the water hammer effect, which is in turn their main drawback.*

Author response: (marked up manuscript lines 33-35)

The automatic hydraulic ram has been used for centuries to lift water to heights over 100 meters and was considered an effective and highly reliable machine for pumping water once certain conditions are
545 satisfied.

***Referee comment:*** *L. 41: It mentions "water was wasted", whereas the Fig. 1 refers to "exhausted water". It is important to keep consistency in the nomenclature, and making sure it matches with the usual terminology in the literature (e.g. "A Manual On The Hydraulic Ram For Pumping Water" by S.B. Watt,*
550 *or "Hydraulic Ram Pumps: A guide to ram pump water supply systems" by Jeffrey et al.)*

Author response:

In the marked up manuscript version; the word waste water was replaced with exhausted water throughout the manuscript.

***Referee comment:*** *L. 43: About "once any form of flow can be created", it will be good to provide an*
555 *insight on how this flow can be created after the different water sources, i.e. what kind of extra infrastructure it might need: dam, weir, drop, etc.*

Author response: (marked up manuscript lines 38-40)

This source could be a spring, streams, river, ponds, dam, lakes and even some wells, once the conditions
560 existed for these water sources to create a hydraulic flow head, either by forming a dam or a naturally existing head.

***Referee comment:*** *L. 44: The ram pump installed at a lower location than the water source is not an ideal scenario but a mandatory one. Otherwise the pump will not operate whatsoever.*

565 Author response: (marked up manuscript lines 41-43)

The ram pump must be installed at a location lower than the water source which was used to create the flow giving the fluid (water) some velocity.

*__Referee comment:__* *L. 48-41: This paragraph describes the generic structure of a Hydram. I suggest to*
570 *put that in the introduction, so in this section the specific parts and assembly methods of your prototype*
*are directly described.*

Author response: In my opinion, I would prefer this paragraph to remain as is.

*__Referee comment:__* *L. 61: Figs 1 and 2 could be put side to side, so the reader can make a quick*
*correspondence between the scheme and the actual prototype.*

575 Author response:  This change was made.

*__Referee comment:__*  L. 73-76: I recommend to match the parts of the experimental set-up, as described in
this paragraph, with those of the Fig. 3, to make sure all of them can be identified in both graphic and
text. A good way of achieving it could be by assigning letters or numbers to each part.

580 Author response: (marked up manuscript lines 137-142)

Figure 3 shows a schematic of the experimental apparatus. The experimental set-up for testing the hammer
head pump was designed with a variable head input (a) and an adjustable head output (b).  The water
supply  was from a 5000L water reservoir (c).  The constant head supply tank was
designed with a float (d) that maintained the constant water level as water was supplied to the inlet of the
585 pump. The input head was the difference in height between the inlet of the pump and the water level at
the top of the constant head supply tank. The outlet side of the pump used variable length of 13mm
diameter PVC pipe to adjust the delivery height (b).

Figure 3 (marked up manuscript line 148)

590 *__Referee comment:__* *L. 93: I wonder if it would be more convenient to combine both tables 1 and 2 in a*
*single one, due to their similar structure. In that case, each cell must be divided in two parts, for the*
*pumped flow and wasted flow, respectively. Moreover, this can give the chance to include a third part:*
*the pumped/wasted ratio; it can be eventually related to the pumping efficiency.*

Author response: An attempt was made to combine the tables, however, the data was too much to fit
595 properly side-by-side on one table.  Hence, the tables were left as is.

*Referee comment:* *L. 100 and next ones: The discussion part must be enriched by comparing your study with other ones, perhaps using similar prototypes in sizes and conditions. Of course, the respective literature and references must be taken into account while doing so.*

Author response: (marked up manuscript lines 169-206)

The findings from the study were compared with published literature and the appropriate comparison were made with respect to references [16] to [19] in the discussion section of the marked-up manuscript.

*Referee comment:* *L. 134: The first conclusion might be too obvious, after so many years of continuous and ubiquitous use of the hydrams, to be considered as such after the present study.*

Author response: (marked up manuscript line 209)

This conclusion was removed

> ➢

*Referee comment:* *Keywords: "Pump characteristic" is not a so accurate keyword for this study, since it does not suggest any possible topic*

Author response: (marked up manuscript lines 21-22)

**Keywords:**

P Hammer head, hydraulic ram , water pump.

*Referee comment:* *L. 26: Farmlands are (missing plural).*

Author response: (marked up manuscript line 46)

In developing and under-developed countries, farmlands are

*Referee comment:* *L. 37: The name of the son of Joseph Michel de Montgolfier, who improved his father's model, is Pierre (or Pierre François).; L. 37: "Montgolfier designed the sniffer valve that reintroduce..." It must be "reintroduced"*

Author response: (marked up manuscript line 31)

1816 this problem was eliminated when Pierre Montgolfier designed the sniffer valve that reintroduced

625 ***Referee comment:*** *L. 40: "The pump construction was simple and consisted of a pump camber...". Do you refer to "chamber" perhaps?*

Author response: (marked up manuscript line 32)

air into the chamber.